# Microfluidic Sliding Paper-Based Device for Point-of-Care Determination of Albumin-to-Creatine Ratio in Human Urine

**DOI:** 10.3390/bios12070496

**Published:** 2022-07-07

**Authors:** Szu-Jui Chen, Chin-Chung Tseng, Kuan-Hsun Huang, Yu-Chi Chang, Lung-Ming Fu

**Affiliations:** 1Department of Engineering Science, National Cheng Kung University, Tainan 701, Taiwan; kame9602@gmail.com (S.-J.C.); n96104496@gs.ncku.edu.tw (K.-H.H.); christina780712@gmail.com (Y.-C.C.); 2Division of Nephrology, Department of Internal Medicine, National Cheng Kung University Hospital Dou-Liou Branch, College of Medicine, National Cheng Kung University, Yunlin 640, Taiwan; chinchun@mail.ncku.edu.tw; 3Department of Internal Medicine, National Cheng Kung University Hospital, College of Medicine, National Cheng Kung University, Tainan 701, Taiwan; 4Graduate Institute of Materials Engineering, National Pingtung University of Science and Technology, Pingtung 912, Taiwan

**Keywords:** albumin, creatinine, colorimetric, paper-based

## Abstract

A novel assay platform consisting of a microfluidic sliding double-track paper-based chip and a hand-held Raspberry Pi detection system is proposed for determining the albumin-to-creatine ratio (ACR) in human urine. It is a clinically important parameter and can be used for the early detection of related diseases, such as renal insufficiency. In the proposed method, the sliding layer of the microchip is applied and the sample diffuses through two parallel filtration channels to the reaction/detection areas of the microchip to complete the detection reaction, which is a simple method well suited for self-diagnosis of ACR index in human urine. The RGB (red, green, and blue) value intensity signals of the reaction complexes in these two reaction zones are analyzed by a Raspberry Pi computer to derive the ACR value (ALB and CRE concentrations). It is shown that the G + B value intensity signal is linearly related to the ALB and CRE concentrations with the correlation coefficients of R^2^ = 0.9919 and R^2^ = 0.9923, respectively. It is additionally shown that the ALB and CRE concentration results determined using the proposed method for 23 urine samples were collected from real suffering chronic kidney disease (CKD) patients are in fine agreement with those acquired operating a traditional high-reliability macroscale method. Overall, for point-of-care (POC) CKD diagnosis and monitoring in clinical applications, the results prove that the proposed method offers a convenient, real time, reliable, and low-spending solution for POC CKD diagnosis.

## 1. Introduction

Chronic kidney disease (CKD) affects a gradual loss of normal kidney function and is generally associated with diabetes, high blood pressure, or hypertension. In the early stage of its development, CKD shows very few signs or symptoms. However, as the disease progresses, CKD patients may experience nausea and vomiting, fatigue and sleep problems, shortness of breath, lower back pain, and the swelling of the feet and ankles. In severe cases, CKD may also result in vascular disease, cognitive dysfunction, anemia, osteoporosis, and fractures [1]. CKD is generally diagnosed through blood and urine tests. Human urine consists of around 95% water and a large number of smaller constituents, such as uric acid, ALB, CRE, sodium, potassium, ammonia, nitrogen, magnesium, and others [2,3,4,5,6,7,8]. Urine may also include precipitates of calcium, phosphorus, and other compound crystals [9]. In healthy humans, the albumin content in urine is less than 30 mg/g, and any value higher than this may indicate potential kidney disease [10]. Thus, routine urine examinations are essential to detect possible disorders of the kidney and urinary system. The components of human urine are extremely sensitive to the daily food intake, disease, drugs, and other personal factors [11,12]. Thus, kidney function disorders and kidney disease are commonly diagnosed through 24-h urine tests, in which the urine is gathered over 24 h to obtain a more reliable evaluation of the body condition [13]. However, such tests are not only inconvenient from the patient’s perspective but may also have the risk of low adherence to instructions. Accordingly, effective techniques for diagnosing and monitoring CKD using normal single urine samples are required.

The ACR (albumin-to-creatine ratio) is a significant prognostic marker for CKD patients [14,15] and is widely used as an early indicator of glomerular lesions [16,17,18] and for the on-going monitoring of diabetes and hypertension [19,20]. Notably, the ACR value can be determined from single urine samples and thus provides a more convenient diagnostic tool than 24-h urine tests [21,22]. The CRE in human urine determination is usually performed via a Jaffé kinetic assay, involving alkaline picric acid as the reagent [23,24]. Many techniques are available for CRE concentration determination, including gas chromatography (GC) [25,26,27], liquid chromatography–mass spectrometry (LCMS) [28,29,30], spectrophotometry, colorimetry [31,32], and electrochemical sensors [33,34]. Meanwhile, the ALB concentration in urine is usually decided by monitoring the variation in the absorbance intensity of the sample reagent following the addition of Bromocresol Green (BCG) dye [35,36]. Again, many methods for ALB determination are available, including high-performance liquid chromatography (HPLC) [37,38], capillary electrophoresis [39,40], chemiluminescence [41,42], and light scattering [43]. However, for both ALB and CRE concentrations detection, the associated analytical techniques are complex and time-consuming, involve expensive and bulky instrumentation, have a long reaction time, require large samples under the test and massive sample reagent volumes, and demand the services of skilled medical technicians. As a result, they are impractical for low-cost, POC applications. Consequently, there is an outstanding requirement for cheaper, simpler, and more portable solutions [44,45].

Recent studies have shown that paper-based devices provide a viable method for detecting early CKD by assessing the concentration of ALB, MAU (microalbuminuria), CRE, or ACR in urine samples [4,19,46,47,48,49,50]. For example, Cai et al. [46] presented a field amplification paper-based device for sensitive colorimetric detection of MAU in human urine. The field-amplified stacking (FAS) technology has been integrated into the paper-based device, which uses the geometric shape of the 2D paper-based microfluidic channel to introduce an electric field gradient and achieve field amplification by reduced the sample conductivity. The experimental results show that the MAU concentration reacted linearly in the range of 10–100 mg/L, and the LOD was 6.5 mg/L. Mathaweesansurn et al. [48] produced a contact imprint paper-based device for the multi-point standard addition determination of CRE in human urine. The single-step embossing method using rubber stamps can simultaneously make eight individual hydrophobic barrier structure patterns on a paper-based device for multi-point standard analysis. This detection device also utilizes the method of orange-colored product produced by Jaffé reaction to detect human urine CRE concentration. A linear concentration of urine Cre of 50 to 1000 mg/L was obtained, and the LOQ was 16.9 mg/L. Hiraoka et al. [50] developed a drawing paper-based device for urinary ACR analysis to the early detection of renal insufficiency. In this device, the ACR is determined by drawing a straight line between the tops of the two color-changing analysis channel sections and then observing the position of the intercept point on ALB-to-CRE scale placed between them. The results obtained for 12 urine samples collected from a hospital showed that the detection results for the ACR were within 15% of the theoretical values.

Paper-based analytical devices (μPADs) have a high integration capability and good biocompatibility and have found extensive utilization in various fields in the recent years, such as biomedicine, environmental monitoring, food science, pharmaceuticals, drug development, and more [51,52,53,54,55,56,57,58]. In clinical applications, paper-based devices have many advantages over the benchtop techniques described above, including a significantly lower cost, faster throughput, simpler operation, lower reagent consumption, and greater portability [59,60,61,62]. As a result, they have significant potential for the realization of point-of-care testing (POCT) systems [63,64,65,66,67].

Accordingly, this study proposes a simple assay platform for determining the ALB and CRE concentrations in real human urine using a sliding microfluidic paper-based chip and intrusion detection system with Raspberry Pi. Briefly, 50 μL of real human urine is dripped into the sample reservoir of the paper-based microchip and diffuses to two reaction zones containing BCG and picric acid, respectively. The chip is then heated at 40 °C and held for 3 min to actuate the reaction of colorimetric. The Raspberry Pi with software is analyzed the RGB color value intensity of the two reaction areas and the concentrations of ALB and CRE are derived using simple calibration formulae prepared in advance using control samples with known concentrations. The presented detection platform system is confirmed feasibility by comparing the ALB and CRE detection results for 23 real-world urine samples collected from CKD patients with the results obtained from a certified detection technique. The specimen used in this study is a non-invasive sample of urine that can be easily collected without pain and requiring no special equipment. As a result, it has significant potential for point-of-care testing (POCT) or home health monitoring and diagnosis.

## 2. Materials and Methods

### 2.1. Control Samples and Reagents

ALB and CRE control samples were acquired from National Cheng Kung University Hospital (NCKUH, Tainan, Taiwan) together with the corresponding reagents required for reaction purposes. For ALB detection, small strips of filter paper (Advantec Grade No. 4A Hardened Ashless Filter Paper, TOYO ROSHI KAISHA LTD., Tokyo, Japan) were coated with a reagent solution consisting of BCG dye (6.6 mmol/L) mixed with ethanol (70%) and then diluted to approximately pH 4.1 using PBS buffer (0.1 M). Under appropriate reaction conditions (temperature and time), a reaction complex was thus formed as:(1)$$\mathrm{Albumin}+\mathrm{Bromocresol}\;\mathrm{Green}\;\overset{PBS\;\left(pH4.1\right)}{\to }\;\mathrm{ALB}-\mathrm{BCG}\;\mathrm{Complex}$$

Similarly, to facilitate CRE detection, small strips of filter paper were coated in a mixed reagent solution consisting of 50 mmol PA (Picric Acid) Reagent 2 (pH = 6.5) and 1 mol NaOH Reagent 1 (pH > 13.5). Colorimetric detection was subsequently performed in accordance with the following Jaffé reaction:(2)$$\mathrm{Creatinine}+\mathrm{Picric}\;\mathrm{Acid}\;\overset{NaOH}{\to }\;\mathrm{CRE}-\mathrm{Picrate}\;\mathrm{Complex}$$

### 2.2. Manufacture of Sliding Microfluidic Double-Track Paper-Based Chip

Figure 1a illustrates the basic structure of the sliding microfluidic double-track paper-based chip proposed in the present study. As shown, the microchip consisted of five main layers and had overall dimensions of 50 mm × 20 mm × 4.5 mm. The packaging layer consisted of a *polyethylene terephthalate* (*PET*) layer coated with a gold membrane for aesthetic purposes and incorporating an opening for the sliding actuator. The second layer was also fabricated of PET and consisted of a large open structure with two color bars adhering to the two side branches to facilitate a qualitative analysis of the ALB and CRE concentrations of the urine sample, respectively. The third and fourth layers of the microchip (the sliding layers) were fabricated of *polymethyl methacrylate* (*PMMA*) and contained a sample chamber with dimensions of 5.5 mm × 5 mm, two reaction/detection areas with dimensions of 2.5 mm × 2.5 mm, and two filtration channels with dimensions of 0.3 mm × 5 mm connecting the sample chamber and reaction/detection zones, respectively. Finally, the fifth layer of the microchip was also fabricated of PMMA and served to improve the rigidity of the device and ensure a uniform distribution of the urine sample on the filter paper strips within the two reaction/detection zones.

The various layers within the microchip were designed using AutoCAD (2020) and CorelDRAW (2020) software and were manufactured by the cutting machine with CO_2_ laser (GIANT TECH.INC Co., Ltd., Taipei, Taiwan). In the assembly process, the reagent-coated strips of filter paper were cut to size (2.5 mm × 2.5 mm) and placed in the two reaction/detection zones, and two small pieces of blood separation membrane (F1, Advanced Microdevices Pvt. Ltd., Saha, India) were introduced into the two microfiltration channels. Two PET layers were bonded using an oxygen plasma treatment and were then sealed with the PMMA layers using a thermal compression technique to form the final microchip assembly (see Figure 1b,d).

### 2.3. Hand-Held Detection System

Figure 2 shows the hand-held detection system developed in the present study with dimensions of approximately 150 mm × 75 mm × 40 mm and a weight of around 492 g. As shown, the main components of the system which included a Raspberry Pi computer (Raspberry Pi 4 Model B, Tainan, Taiwan), a Raspberry Pi Camera (Sony IMX219, 8-megapixel, Raspberry Pi, Taiwan Taiwan), a temperature controller (DHT 11, TAIWANIOT, Co., Ltd., Yunlin, Taiwan), a touch switch module (TTP223-BA6, Tainan, Taiwan), a voltage controller (DC-DC, MP1584, Taiwan), two rectangular LED white light sources (CL-S257W8, ShenZhen Caijing, Co., Ltd., Shenzhen, China), and a lithium battery (3.7 V, 4000 mAh, 606090, Tainan, Taiwan).

Note that two (rather than one) LEDs were installed in the detection box in order to improve the illumination uniformity of the reaction zone and avoid light halo effects. The system also incorporated an insertion slot for the sliding microchip and a Raspberry Pi smart tablet (Raspberry Pi 3.5″, Taiwan) mounted on top of the detection box (not shown) for visualizing the ALB and CRE concentrations detection results.

### 2.4. ALB and CRE Concentrations Detection Process

In the detection process, the sliding PMMA element in the proposed microchip was extended and 50 μL of real human urine was dripped into the sample reservoir. The sliding element was then retracted, and the urine sample was allowed to diffuse naturally through the filtration channels until it reached the two reaction/detection zones. Once the filter paper strips in the reaction/detection zones were fully wetted (after approximately 10 s), the microchip was inserted into the detection box and the temperature set to 40 °C for heating, the chip was left for 3 min to actuate the colorimetric reaction in both reaction/detection zones. The complex compounds formed in the two detection areas were observed by the Raspberry Pi camera and the resulting images were interfaced to the Raspberry Pi computer. The RGB color intensities of the two compounds (ALB + BCG and CRE + Picrate) were then analyzed and the ALB and CRE concentrations were determined using calibration formulae prepared in advance using ALB and CRE control samples with known concentrations. Finally, the ALB and CRE concentrations were displayed on the Raspberry Pi tablet on top of the detection box.

Notably, the use of a sliding chip design minimizes the effects of evaporation during the sample filtration and colorimetric reaction stages and therefore improves the RGB intensity value of the colorimetric images and enhances the detection performance as a result. Furthermore, in contrast to traditional benchtop analytic techniques, the proposed microchip and detection box enable the ALB and CRE concentrations of the urine sample to be determined simultaneously in a single detection process. As a result, the presented detection system has substantial potential for low-spending POCT applications in clinical CKD practice.

In the present study, the ALB and CRE concentrations are determined by the Raspberry Pi computer based on an inspection of the RGB intensity signal. However, as shown in Figure 1a, the microchip also incorporates two color bars adjacent to the reaction/detection zones to facilitate a qualitative determination of the ALB and CRE concentrations. It is thus anticipated that the sliding microchip can also be used as a crude standalone system for CKD diagnosis without the need for the full functionality of the Raspberry Pi-based detection system (e.g., a simple hotplate is sufficient).

## 3. Results and Discussion

### 3.1. Optimization of ALB and CRE Reaction Conditions

According to the findings of previous studies by the present group [24], the colorimetric reaction within the detection box was performed at the 40 °C for the waiting time of 3 min. Moreover, the G + B value intensity signal of the detection areas was found to be correlated mostly strongly with the ALB and CRE concentrations of the urine sample [35,36]. However, the RGB intensity of the reaction complexes also depends on the NaOH concentration (CRE reaction) and BCG concentration (ALB reaction). Accordingly, an experimental investigation was performed to determine the optimal NaOH and BCG concentrations using ALB and CRE control samples with known concentrations in the range of 10–300 mg/dL and 0.75–10 mg/dL, respectively.

In previous studies [24,35,36,68], the optimal reaction temperature for the detection of ALB and CRE by a paper-based device was mostly set to 37 °C. However, the current study used a slip-hybrid PMMA/paper microchip to integrate ALB and CRE detection on this microchip. Therefore, the optimal reaction temperature for this study was set at 40 °C to allow efficient and continuous heating from PMMA to paper-based components. In the optimal reaction time, the optimal reaction time was set to 3 min. For a more detailed description of the optimal reaction temperature and reaction time, please refer to our previous studies [24,35,36].

Figure 3 shows the G + B intensity value variation with the CRE concentration for four distinct values of the NaOH concentration (1–4 M) in the NaOH/picric acid reagent. For all values of the CRE concentration, the G + B intensity value reduces with a raising NaOH concentration, owing to the lower solubility of picric acid in more alkaline environments. Moreover, for each NaOH concentration, the G + B intensity value reduces approximately linearly with an increasing CRE concentration. However, regression analysis inspection results show that the reagent with a NaOH concentration of 3 M yields the highest correlation coefficient (R^2^ = 0.9923) between the G + B intensity value and the CRE concentration. Thus, the concentration of NaOH was selected as 3 M in all of the remaining experiments.

Figure 4 shows the G + B intensity value variation with the ALB concentration for four distinct values of the BCG concentration (1.155∼2.64 mmol/L). Of the various reagents, that with a BCG concentration of 1.196 mmol/L yields both the highest sensitivity and the highest correlation coefficient (R^2^ = 0.9919). Thus, the BCG concentration for the ALB reagent was set as 1.196 mmol/L in all of the remaining determination experiments.

### 3.2. Calibration Equations for ALB and CRE Detection

The proposed assay platform was calibrated using ALB control samples with known concentrations of 0.75, 2, 4, 6, and 10 mg/dL, respectively, and CRE control samples with concentrations of 10, 50, 100, 200, and 300 mg/dL, respectively. For all of the samples, the G + B intensity value were acquired under the optimal reaction conditions described in the previous section. Figure 5 and Figure 6 present the mean G + B intensity values obtained for the ALB and CRE control samples, respectively, over five repeated measurements in each case. The regression analysis results in Figure 5 indicate that the G + B intensity value (Y) is related to the ALB concentration (X) as Y = −10.932X + 405.418 with a correlation coefficient of R^2^ = 0.9919. Similarly, in Figure 6, the G + B intensity is related to the CRE concentration as Y = −0.414X + 336.192 with a correlation coefficient of R^2^ = 0.9923. In both cases, the correlation coefficients have a high value close to 1. In other words, the calibration curves provide a reliable means of predicting the ALB and CRE concentrations from the measured G + B intensity values.

### 3.3. Application of Proposed Assay Platform to Real-World Urine Samples

Human urine samples were collected from 23 adult CKD patient volunteers at National Cheng Kung University Hospital (NCKUH, Taiwan). For each sample, the hospital provided the age and gender of the patient, the ALB and CRE concentrations, and the corresponding ACR value (see Table 1). Note that the ALB and CRE concentrations were determined using a conventional method on a benchtop biochemistry analyzer (Hitachi-7600, Hitachi High-Technologies Co., Minato, Japan).

The ALB and CRE concentrations of each sample were determined using the proposed assay platform under the optimal reaction conditions described in Section 3.1 and the calibration equations presented in Section 3.2. The corresponding results are shown in Figure 7a,b, where the Y-axis in each figure shows the benchmark results provided by NCKUH and the X-axis indicates the measurement results obtained using the proposed paper-based platform. It can be seen that a good agreement exists between the two sets of measurements in both cases (i.e., R^2^ = 0.9933 for ALB and R^2^ = 0.9980 for CRE). Thus, the feasibility of the proposed platform for practical ALB and CRE determination applications is confirmed. Table 2 presents the comparison of several properties between the current platform detection system and the developed methods for ALB and CRE detection in urine samples.

## 4. Conclusions

This study has presented an effective and low-cost approach for determining the ALB and CRE concentrations in single human urine samples using a simple sliding microfluidic paper-based device and a hand-held detection system based on RGB color intensity analysis. Notably, the ALB and CRE concentrations can be detected through a single colorimetric reaction conducted at a low temperature of 40 °C for just 3 min. The experimental results have shown that, given an optimal composition of the ALB and CRE reagents, the G + B intensity value of the reaction compound varies linearly (R^2^ = 0.9919) with the ALB concentration over the range of 0.75–10 mg/dL. Similarly, the G + B intensity varies linearly (R^2^ = 0.9923) with the CRE concentration over the range of 10–300 mg/dL. Furthermore, the ALB and CRE concentration measurements obtained for the urine samples of 23 CKD patient volunteers are in excellent agreement (R^2^ = 0.9933 and R^2^ = 0.9980, respectively) with the benchmark values obtained using a conventional macroscale technique.

The results indicate that the proposed assay platform provides a feasible alternative to conventional benchtop techniques for determining the ALB and CRE concentrations in human urine samples and computing the ACR accordingly. As such, it provides a feasible solution for the low-cost, rapid, and straightforward detection of CKD in both clinical and POC contexts. In the present study, the ALB and CRE concentrations are derived from a computer-based analysis of the RGB intensity of the reaction compounds. However, the microchip also incorporates two color bars which enable a crude assessment of the ALB and CRE concentrations to be made through a simple naked-eye inspection. Thus, the potential of the proposed microchip for preliminary CKD diagnosis in non-clinical contexts or regions of the world with poorly-developed medical infrastructures is further enhanced.

## Figures and Tables

**Figure 1 biosensors-12-00496-f001:**
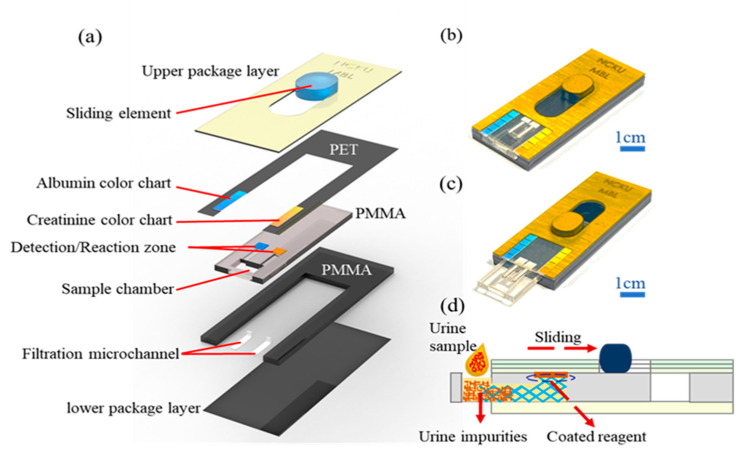
(**a**) Detailed structural arrangement of sliding microchip for simultaneous ALB and CRE detection. (**b**,**c**) Schematic representations of sliding microchip in retracted and extended conditions. (**d**) Cross-sectional view of sliding microchip in retracted condition.

**Figure 2 biosensors-12-00496-f002:**
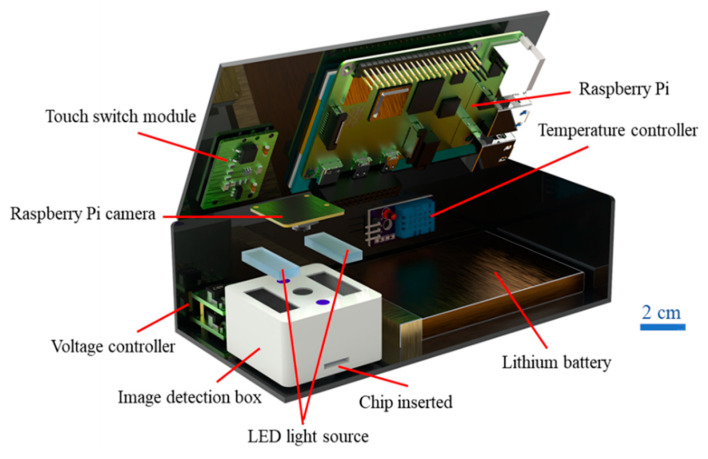
Main components of hand-held detection system.

**Figure 3 biosensors-12-00496-f003:**
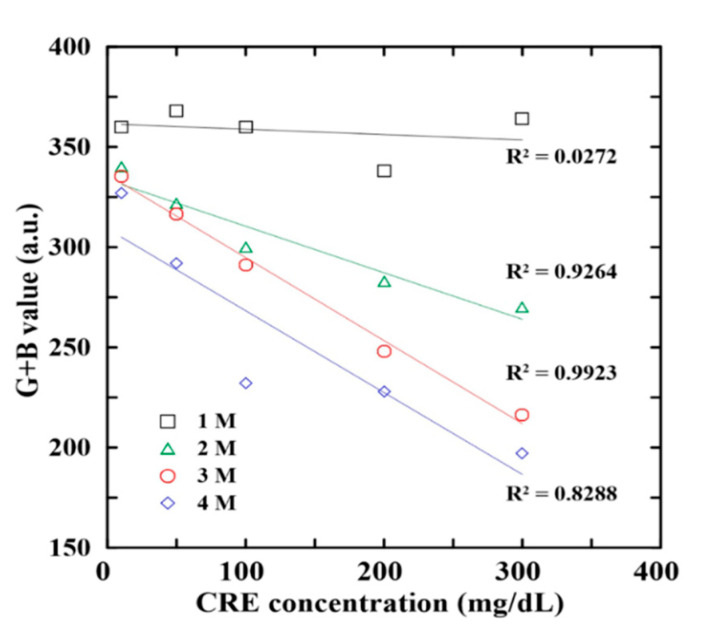
Effects of NaOH concentration on G + B intensity value of reaction complexes with different CRE contents.

**Figure 4 biosensors-12-00496-f004:**
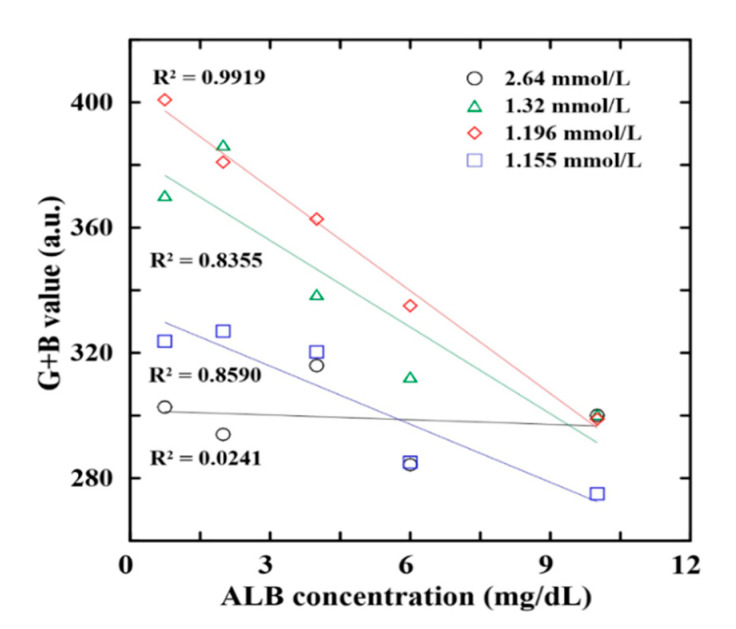
Effects of BCG concentration on G + B intensity value of reaction complexes with four different ALB contents.

**Figure 5 biosensors-12-00496-f005:**
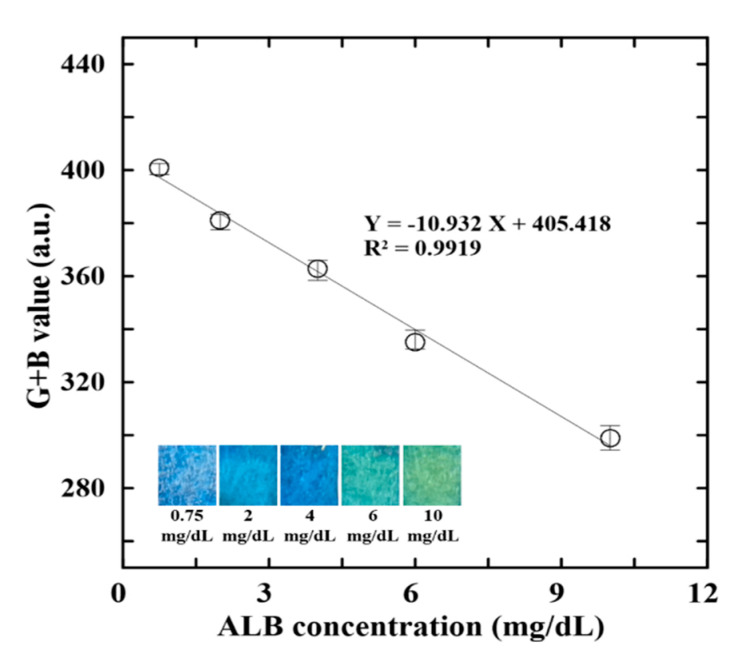
Variation of G + B intensity value with ALB concentration.

**Figure 6 biosensors-12-00496-f006:**
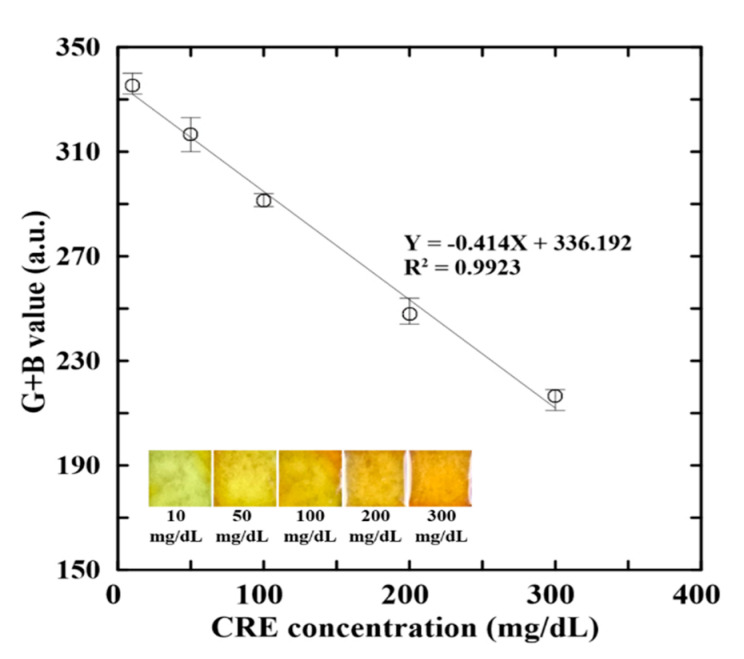
Variation of G + B intensity value with CRE concentration.

**Figure 7 biosensors-12-00496-f007:**
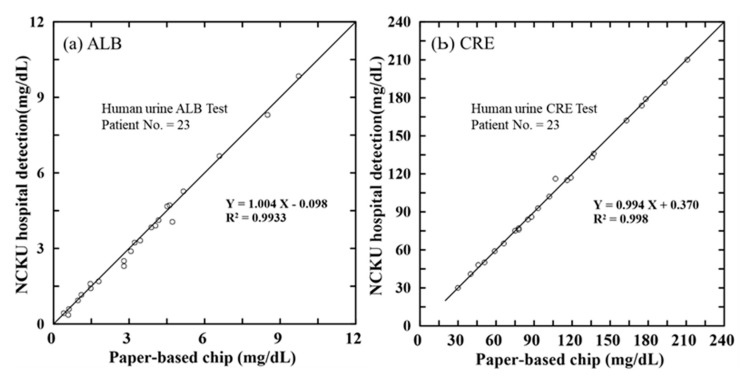
Comparison of benchtop results and paper-based chip results for: (**a**) ALB concentration and (**b**) CRE concentration of real-world urine samples.

**Table 1 biosensors-12-00496-t001:** Benchmark ACR values of urine samples collected from real-world CKD patients of different genders and ages.

Gender	Age	Number	<10 mg/dL	10–300 mg/dL	>300 mg/dL
**Male**	All	19	4	8	7
	<40	3	0	2	1
	40-60	3	1	1	1
	>60	13	3	5	5
**Female**	All	4	1	1	2
	<40	2	1	0	1
	40-60	1	0	1	0
	>60	1	0	0	1

**Table 2 biosensors-12-00496-t002:** Comparison of several properties between the current platform detection system and the developed methods for ALB and CRE detection in urine samples.

	Sequential Injection Analysis [69]	Smartphone EC Method [70]	Microfluidic Chip Method [68]	Paper-Based Colorimetry [21]	Current Platform
**Sample type**	Urine	Artificial urine	Urine	Urine	Urine
**Detection method**	SP-metric	EC	Optical detection	Colorimetric	Colorimetric
**Sample consumption**	100 μL	100 mL	6 μL	2 μL	50 μL
**Analysis time**	1 min	3 min	1 min	30 s	3 min
**Detection range**	2–20 mg/L (ALB)	10–2000 μg/mL (ALB)	5–220 mg/L (ALB)	10–350 mg/dL (ALB)	0.75–10 mg/d (ALB)
	5–100 mg/L (CRE)	10–300 μg/mL (CRE)	1–100 mg/L (CRE)	10–400 mg/dL (CRE)	10–300 mg/dL (CRE)
**Operating temperature**	NA	NA	37 °C	NA (ALB)	40 °C
				50 °C (CRE)	
**Reagent or medium**	Eosin Y	BSA	Non	BCG,	BCG,
	Picric acid	Picric acid	immunological fluorescent	Picric acid	Picric acid
**Price device**	High	High	High	Low	Low
**Instrument size**	Benchtop	Hand-held	Benchtop	Hand-held	Hand-held
**LOD**	0.06 mg/dL (ALB)	6.3 mg/dL (ALB)	0.14 mg/dL (ALB)	7.1 mg/dL (ALB)	0.5 mg/dL (ALB)
	3.5 mg/dL (CRE)	9.3 mg/dL (CRE)	0.4 mg/dL (CRE)	5.4 mg/dL (CRE)	5 mg/dL (CRE)
**Real sample**	39	0	40	0	23

EC: Electrochemical; SP-metric: Spectrophotometric.

## Data Availability

The data presented in this study are available upon request from the corresponding author.

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
