# Peer review of "Microfluidic Sliding Paper-Based Device for Point-of-Care Determination of Albumin-to-Creatine Ratio in Human Urine"

_biosensors, 2022, doi:10.3390/bios12070496_

Round 1

Reviewer 1 Report

The peer-reviewed manuscript describes an interesting platform for performing, at the point of care, the albumin to creatine ratio tests from real human urine. The manuscript is interesting and well written, but some parts need to be improved, especially the scientific value of the article should be increased. Now, the article is more like a “technical note” than a full-size scientific paper. I would recommend expanding the part presenting the research results or specifying the title/purpose of the article.
Regardless, the article may be published with minor corrections.

Detailed comments:
1. Abstract, lines 17-27 present the operating protocol of the platform rather than describes the scientific content of the article. It should be corrected.

2. Introduction, although interesting, does not present the current status of albumin/creatine microfluidic test device development - the additional half-page review is highly desirable. The authors should also discuss the originality of the presented solution in the context of their previous work. Is this the first article on this topic, or has the described platform already been used in research? It would be very good if the introduction ended with the formulation of a scientific thesis.

3. Conclusions. Would the authors be able to compare the obtained test results with the results presented in other scientific publications (competing solutions)?

Author Response

Reviewer #1:
The peer-reviewed manuscript describes an interesting platform for performing, at the point of care, the albumin to creatine ratio tests from real human urine. The manuscript is interesting and well written, but some parts need to be improved, especially the scientific value of the article should be increased. Now, the article is more like a “technical note” than a full-size scientific paper. I would recommend expanding the part presenting the research results or specifying the title/purpose of the article.
Regardless, the article may be published with minor corrections.

Detailed comments:
1. Abstract, lines 17-27 present the operating protocol of the platform rather than describes the scientific content of the article. It should be corrected.
Reply: Thanks to the reviewer. Authors revised the “Abstract”. (Page 1)

  1. Introduction, although interesting, does not present the current status of albumin/creatine microfluidic test device development - the additional half-page review is highly desirable. The authors should also discuss the originality of the presented solution in the context of their previous work. Is this the first article on this topic, or has the described platform already been used in research? It would be very good if the introduction ended with the formulation of a scientific thesis.

Reply: Thanks to the reviewer. Authors added some references and a paragraph to illustrate the current state of the ALB/CRE microfluidic paper-based analysis devices, and revised the Introduction section. (Pages 2-3)

  1. Conclusions. Would the authors be able to compare the obtained test results with the results presented in other scientific publications (competing solutions)?

Reply: Thanks to the reviewer. Authors have added some references and a table (Table 2) to compare existing technologies. (Page8- 9)

Reviewer 2 Report

In this manuscript, the authors exploited a portable platform consisting of a microfluidic sliding double-track paper-based chip and a hand-held detection system for analyzing the albumin to creatine ratio. They investigated the reaction conditions by determining the optimal NaOH and BCG concentrations and demonstrated a reliable method of predicting the ALB and CRE concentrations from the measured G + B intensity values. This work offers a feasible alternative to conventional benchtop techniques for determining the ALB and CRE concentrations in human urine samples. However, I have several minor concerns before this manuscript can be accepted. Therefore, in its current form, revisions are needed.

1. The introduction part is incomplete, since they only introduced the advantages of paper based devices. The significance of this work should be elaborated comprehensively. For example, the background and progress of paper based chips for colorimetrically detecting diseases, especially ALB and CRE, should be given.

2. The authors investigated the influence of ALB and CRE reaction conditions by optimizing the NaOH concentration (CRE reaction) and BCG concentration (ALB reaction). Did the authors study other experimental conditions? For instance, how will the operation time, the concentration and volume of reagents coated in the filter paper influence the colorimetric detection results?

Author Response

Reviewer #2:
In this manuscript, the authors exploited a portable platform consisting of a microfluidic sliding double-track paper-based chip and a hand-held detection system for analyzing the albumin to creatine ratio. They investigated the reaction conditions by determining the optimal NaOH and BCG concentrations and demonstrated a reliable method of predicting the ALB and CRE concentrations from the measured G + B intensity values. This work offers a feasible alternative to conventional benchtop techniques for determining the ALB and CRE concentrations in human urine samples. However, I have several minor concerns before this manuscript can be accepted. Therefore, in its current form, revisions are needed.

  1. The introduction part is incomplete, since they only introduced the advantages of paper based devices. The significance of this work should be elaborated comprehensively. For example, the background and progress of paper based chips for colorimetrically detecting diseases, especially ALB and CRE, should be given.

 Reply: Thanks to the reviewer. Authors added some references and a paragraph to illustrate the current state of the ALB/CRE microfluidic paper-based analysis devices, and revised the Introduction section. (Pages 2-3)

  1. The authors investigated the influence of ALB and CRE reaction conditions by optimizing the NaOH concentration (CRE reaction) and BCG concentration (ALB reaction). Did the authors study other experimental conditions? For instance, how will the operation time, the concentration and volume of reagents coated in the filter paper influence the colorimetric detection results?

 Reply: Thanks to the reviewer. Authors add a paragraph to explain the optimal operating conditions of the current platform. (Page 6)

“In previous studies [24,35,36,68], the optimal reaction temperature for detection of ALB and CRE by paper-based device was mostly set to 37oC. However, the current study used a slip-hybrid PMMA/paper microchip to integrate ALB and CRE detection on this microchip. Therefore, the optimal reaction temperature for this study was set at 40oC to allow efficient and continuous heating from PMMA to paper-based components. In the optimal reaction time, the optimal reaction time was set to 3 minutes. For a more detailed description of the optimal reaction temperature and reaction time, please refer to our previous studies [24,35,36].”

Reviewer 3 Report

I have reviewed a great interdisciplinary work entitled “Microfluidic Sliding Paper-Based Device for Point-of-Care Determination of Albumin-to-Creatine Ratio in Human Urine” that has been done by authors. They have used microfluidics devices to detection of ALB and CRE concentrations in real human urine. This manuscript is acceptable for publishing in biosensor mdpi,

I have some suggestion to be considered by the authors and after that is ready to publish:

1. In the abstract there is no quantitative value, and overall, in the main text please clarify the limit of detection, if it was done. For example, ACR concentration range and LOD.

2.  Did you have evaluated your data and results by existing commercial techniques? I think if you will carrying out this section your data accuracy will be enhanced.

3. I am a scientist and always I know what is PET, and PMMA, however, you should insert an abbreviation section, and/or clarify what is PET and …that are presented for the first time.

4. Why you have used the microfluidics term? It was not better to use lateral flow paper-based assay instead of microfluidics, your platform is working by LFA.

5. In the case of quantitative results, what is the superior ability of your work in comparison to commercially available techniques? 

Author Response

Reviewer #3:

I have reviewed a great interdisciplinary work entitled “Microfluidic Sliding Paper-Based Device for Point-of-Care Determination of Albumin-to-Creatine Ratio in Human Urine” that has been done by authors. They have used microfluidics devices to detection of ALB and CRE concentrations in real human urine. This manuscript is acceptable for publishing in biosensor mdpi, I have some suggestion to be considered by the authors and after that is ready to publish:

  1. In the abstract there is no quantitative value, and overall, in the main text please clarify the limit of detection, if it was done. For example, ACR concentration range and LOD.

 Reply: Thanks to the reviewer. Usually the ACR concentration range and LOD are determined by detecting ALB and CRE. Authors have added Table 2 to illustrate the detection concentration range and LOD of ALB and CRE. (Pages 8-9)

  1. Did you have evaluated your data and results by existing commercial techniques? I think if you will carrying out this section your data accuracy will be enhanced.

 Reply: Thanks to the reviewer. In Section 3.3, authors describe a commercially available technique used by the National Cheng Kung University Hospital for the detection of ALB and CRE concentrations in urine. (Page 7) At the same time, authors have added some references and a table (Table 2) to compare existing technologies. (Pages 8-9).

“Human urine samples were collected from 23 adult CKD patient volunteers at National Cheng Kung University Hospital (NCKUH. Taiwan). For each sample, the hospital provided the age and gender of the patient, the ALB and CRE concentrations, and the corresponding ACR value (see Table 1). (Note that the ALB and CRE concentrations were determined using a conventional method on a benchtop biochemistry analyzer (Hitachi-7600, Hitachi High-Technologies Co., Japan).)” (Page 8)

  1. I am a scientist and always I know what is PET, and PMMA, however, you should insert an abbreviation section, and/or clarify what is PET and …that are presented for the first time.

 Reply: Thanks to the reviewer. The authors have explained PET and PMMA in the text. (Page 3)

  1. Why you have used the microfluidics term? It was not better to use lateral flow paper-based assay instead of microfluidics, your platform is working by LFA.

 Reply: Thanks to the reviewer. Initially, the authors also considered using LFA instead of microfluidics, but the microfluidic device in this study includes microchannels on a PMMA chip that can be used for microfluidic delivery, so the authors still use microfluidic as the device name.

  1. In the case of quantitative results, what is the superior ability of your work in comparison to commercially available techniques?

 Reply: Thanks to the reviewer. Compared with the commercial technology (National Cheng Kung University Hospital, Taiwan), the current microfluidic detection platform requires a small sample volume, fast response time, and detection results similar to the sensitivity of the commercial technology. The specimen used in this study is a non-invasive sample of urine that can be easily collected without pain and requiring no special equipment. As a result, it has significant potential for point-of-care testing (POCT) or home health monitoring and diagnosis.

Round 2

Reviewer 2 Report

The authors have addressed all of my concerns with the original manuscript. The revised manuscript is ready for publication.